# Optimal treatment strategies for stage I non-small cell lung cancer in veterans with pulmonary and cardiac comorbidities

Keith Sigel [1‡]*, Chung Yin Kong [1‡], Sadiq Rehmani[1], Susan Bates[2,3], Michael Gould[4], Kimberly Stone[1], Minal Kale[1], Yeun-Hee Park[2,3], Kristina Crothers[5,6], Faiz Bhora[7], Juan Wisnivesky[1]

1 Division of General Internal Medicine, Icahn School of Medicine at Mount Sinai, New York, New York, United States of America, 2 James J. Peters VA Medical Center, Bronx, New York, New York, United States of America, 3 Columbia University School of Medicine, New York, New York, United States of America, 4 Kaiser Permanente Southern California, Los Angeles, California, United States of America, 5 University of Washington School of Medicine, Seattle, Washington, United States of America, 6 Puget Sound VA Medical Center, Seattle, Washington, United States of America, 7 Nuvance Health, Danbury, Connecticut, United States of America

‡ Drs. Sigel and Kong are co–first authors, with equal contribution to this article.
* keith.sigel@mssm.edu

**Data Availability Statement:** The data used to create the simulation model in this study is legally protected Veterans Health Administration data that cannot be shared under any circumstances. Only

## Abstract

### Background

Veterans are at increased risk of lung cancer and many have comorbidities such as chronic obstructive pulmonary disease (COPD) and coronary artery disease (CAD). We used simulation modeling to assess projected outcomes associated with different management strategies of Veterans with stage I non-small cell lung cancer (NSCLC) with COPD and/or CAD.

### Patients and methods

Using data from a cohort of 14,029 Veterans (years 2000–2015) with NSCLC we extended a well-validated mathematical model of lung cancer to represent the management and outcomes of Veterans with stage I NSCLC with COPD, with or without comorbid CAD. We simulated multiple randomized trials to compare treatment with lobectomy, limited resection, or stereotactic body radiation therapy (SBRT). Model output estimated expected quality adjusted life years (QALY) of Veterans with stage I NSCLC according to age, tumor size, histologic subtype, COPD severity and CAD diagnosis.

### Results

For Veterans <70 years old lobectomy was associated with greater projected quality-adjusted life expectancy regardless of comorbidity status. For most combinations of tumors and comorbidity profiles there was no dominant treatment for Veterans ≥80 years of age, but less invasive treatments were often superior to lobectomy. Dominant treatment choices differed by CAD status for older patients in a third of scenarios, but not for patients <70 years old.

aggregate data, as reported in this paper can be shared due to legal protections on Veterans health data. The data used in this project was under a waiver of informed consent granted by the Bronx VA Institutional Review Board and therefore even deidentified individual level data cannot be shared. https://www.va.gov/ORO/Docs/Guidance/VA_RSCH_DATA_ACCESS_PLAN_07_23_2015.pdf The source data can be accessed in collaboration with VA researchers, free of charge with appropriate VA IRB approvals. Those wishing to access the raw data that were used for this analysis may contact Keith Sigel, MD PhD (keith.sigel@mssm.edu) or the official VA resource center for research data access (virec@va.gov) to discuss the details of the VA data access approval process. We have confirmed that an interested researcher would be able to obtain a de-identified dataset upon collaborative request pending ethical approval.

**Funding:** This work was supported by the Department of Defense (GRANT W81XWH-16-1-0356, LC150146 to JPW). The funding source had no role in the design, conduct, or analysis of the study or in the decision to submit the manuscript for publication.

**Competing interests:** Dr. Wisnivesky has received consulting honorarium from Sanofi, Glaxosmithkline and Banook and research grants from Sanofi and Quorum. This does not alter our adherence to PLOS ONE policies on sharing data and materials.

## Conclusions

The harm/benefit ratio of treatments for stage I NSCLC among Veterans may vary according to COPD severity and the presence of CAD. This information can be used to direct future research study design for Veterans with stage I lung cancer and COPD and/or CAD.

## Introduction

Lung cancer incidence in Veterans is significantly higher than in the general population [1], due in large part to higher rates of heavy smoking associated with past military service [2]. Localized (stage I) non-small cell lung cancers (NSCLC) account for approximately 23% of cases [3, 4]. However, a considerable increase in the number of Veterans diagnosed with localized disease is expected due to the uptake of lung cancer screening with low dose computed tomography across the Veterans Health Administration (VA) system [5].

Lung cancer patients are generally older (mean age at diagnosis is 70 years) and have multiple comorbidities [6, 7]. Veterans have especially high rates of smoking-related diseases, with 30% of the Veterans qualifying for lung cancer screening reporting at least 2 significant comorbidities [8]. Chronic obstructive pulmonary disease (COPD; 25–50%) and cardiovascular disease (25%) are the most common conditions among lung cancer patients [6, 7]. Stage I lung cancer patients with comorbid illness have lower rates of treatment, increased rates of treatment-related complications, and lower survival [7, 9].

Surgery is the recommended treatment for stage I NSCLC, although there is uncertainty regarding the extent of lung resection (lobectomy or sublobar resection), particularly for patients with comorbidities [10, 11]. Lobectomy is generally the standard of care, especially for tumors greater than 2 cm while limited resection (i.e., wedge resection or segmentectomy) is frequently used for patients with borderline lung function or those at high operative risk. Stereotactic body radiotherapy (SBRT) has emerged as a non-surgical alternative for stage I patients deemed high surgical risk. However, while some data suggest that both limited resection (particularly wedge resection) and SBRT are associated with increased risk of local cancer recurrence, some comparative studies have not found differences [11–15].

Clinical trials of cancer therapies have focused on younger lung cancer patients, largely without major comorbid illness [16]. The results of these trials are unlikely to be directly applicable to the subset of Veterans with serious comorbidities due to increased risks of treatment complications [17, 18] and a greater impact of competing risks (non-lung cancer deaths) [19], as well as lower quality of life, all of which are associated with decreased long-term benefits of aggressive treatments. Optimal lung cancer treatment pathways for stage I NSCLC in Veterans with major comorbid illnesses are therefore unclear leading to challenges in clinical decision-making. To address these uncertainties, we used contemporary national VA data to enhance a well-validated mathematical model of lung cancer to represent Veterans with comorbidities. We then simulated comparative trials of Veterans with stage I NSCLC and COPD, with or without comorbid CAD, to project simulated outcomes for different therapeutic options, thereby estimating the benefits and harms of lobectomy, limited surgical lung resection and SBRT.

## Materials and methods

### Overview of the simulation model

In this study, we extensively reviewed relevant literature and conducted primary data analysis using Veteran data to determine unique factors impacting treatment and outcomes of lung cancer in Veterans with major comorbidities. To synthesize data from these multiple sources

we developed a special treatment-focused version of the Lung Cancer Policy Model (LCPM) [20], a well-validated, comprehensive simulation model of lung cancer development, progression, detection, treatment, and survival.

The modified version of the LCPM focused on the optimal treatment selection for patients with NSCLC, which we refer as the LCPM-Treatment. The LCPM-Treatment incorporated additional new data related to complications and survival, cancer characteristics, and patient comorbidity derived from several national VA sources.

## Lung cancer treatment efficacy

In the LCPM-Treatment, lung cancer treatments lead to lung cancer-specific survival rates according to published information (Table 1). Lobectomy was considered superior to limited resection based on the results of a previous randomized trial of patients with few comorbidities [13]. However, we also incorporated data from observational data showing different efficacy of lobectomy versus limited resection for different tumor sizes [21, 22].

Randomized trials of SBRT versus surgery have been limited by low enrollment and have therefore been inadequately powered to compare these modalities [23]. Therefore, we used observational data from a large VA study using causal inference methods [24] and population-based data estimating the comparative effectiveness of SBRT versus limited resection according to tumor size [25].

## Input data

The input parameters of the LCPM-Treatment include survival, complications and utilities. Parameters for survival, complications and some quality of life conditions were determined from primary VA data obtained from the Corporate Datawarehouse (CDW) and the Veterans Aging Cohort Study (VACS). All other parameters were obtained from review of relevant literature. The data sources used are described in Table 1.

**Lung cancer-specific and non-lung cancer mortality.** We analyzed VA data to estimate non-lung cancer mortality according to age, COPD stage, presence of CAD, and tumor histologic subtype among Veterans with stage I lung cancer. Using cancer registry data from the Oncology Raw domain in the VA CDW we identified 14,029 patients with stage I-IIIA NSCLC (2000–2015) retaining 1,853 subjects who received lobectomy, had available spirometry data (for determination of COPD stage according to degree of impairment in FEV1 in those with airflow obstruction), and cause of death information, which we classified as lung cancer or non-lung cancer-related. We identified COPD and CAD using relevant diagnostic codes (COPD: 490.X-496.X; CAD: 410.X0, 410.X1, 414.X, 412.X) present in the inpatient and outpatient diagnosis files from the 12-months prior to cancer diagnosis. We then fit a Fine-Grey competing risks regression model to estimate baseline cumulative incidence of lung cancer and non-lung cancer death with subhazard ratio estimates for input parameters of interest for each outcome. Use of all data was approved by the Institutional Review Board of the James J. Peters Veterans Administration Medical Center.

**Surgical complications.** To calculate the probability of major surgical complications after lung cancer resection we used data from the VA Surgical Quality Improvement Program (VASQIP), a prospective registry of short-term surgical adverse outcomes, linked to VA cancer registry and clinical data. Using VASQIP we identified common major surgical complications (e.g., 30-day mortality, cerebrovascular accident, myocardial infarction (MI), pneumonia, and others; S1 and S2 Tables) for 6,022 Veterans (and subset of 1,647 with available spirometry data) who underwent lung resection surgery for cancer between 2000–2015. We then fitted logistic regression models that included model input parameters (age, CAD, COPD stage,

**Table 1. Key input parameters for developing a microsimulation of veterans with lung cancer.**

| OBSERVED DATA | DEFINITION | VALUES | | SOURCES |
|---|---|---|---|---|
| **Lung Cancer Treatment Response** | Treatment-specific lung cancer specific survival hazard ratios | **Comparison** | **Hazard Ratio** | |
| | | 0–1 cm LR vs SBRT | 1.16 (0.90–1.53) | [24, 25] |
| | | 1–2 cm LR vs SBRT | 1.16 (0.90–1.53) | [24, 25] |
| | | 2–3 cm LR vs SBRT | 1.16 (0.90–1.53) | [24, 25] |
| | | >3 cm LR vs SBRT | 1.36 (0.98–1.89) | [24, 25] |
| | | 0–1 cm LR vs Lobectomy | 1.24 (0.95–1.61) | [13, 21] |
| | | 1–2 cm LR vs Lobectomy | 1.39 (0.97–2.01) | [13, 26] |
| | | 2–3 cm LR vs Lobectomy | 1.39 (0.97–2.01) | [13, 26] |
| | | >3 cm LR vs Lobectomy | 1.39 (0.97–2.01) | [13, 26] |
| **Non-Lung Cancer Mortality** | Non-lung cancer death rates according to age and comorbidity | | | Primary Data; Appendix Table 1 |
| **Lung Cancer Treatment Complications** | Probability of lung cancer treatment complications including perioperative mortality | | | Primary Data; Appendix Table 2 |
| **Quality of Life for Veterans with COPD / CAD** | Predicted quality of life for Veterans according to comorbidity | **Parameter** | **Utility** | Primary Data; Appendix Table 3 |
| | | Baseline | 0.75 | |
| | | CAD | -0.018 | |
| | | COPD | -0.021 | |
| **Quality of Life Associated with Lung Cancer Treatment Complications** | Disutility associated with major complications of lung cancer treatment | Major Treatment Complication | -0.35* | [27, 28] |

*Quality of life returns to baseline within six months, modeled as a linear recovery.

cancer stage) and use of lobectomy versus limited resection, to generate predictive models for each adverse short-term outcome.

**SBRT complications.** From national VA cancer registry data (2000–2015) in the CDW oncology domain we identified 386 stage I NSCLC patients who were treated with SBRT. Using linked diagnostic code data we identified episodes of serious esophagitis (530.1, 530.10, 530.11, 530.12, 530.19), pneumonitis (508.0, 508.1), and/or hemoptysis (786.3) requiring hospitalization in the 90 days following initiation of SBRT. Given the limited number of adverse outcomes associated with SBRT we did not fit multivariable prediction models but instead used the incidence of these complications as probability estimates in our simulation.

**Quality of life for veterans with COPD and/or CAD.** To estimate quality of life for Veterans with COPD and/or CAD we used baseline short-form 36 (SF-36) data from 3,511 HIV uninfected participants in the prospective VACS collected during 2001–2006 [29]. We implemented a published equation to generate utility scores from SF-36 data [30] and then fitted a linear regression model to predict utility values according to comorbidity status (S4 Table).

**External validation.** For external validation of the simulation we obtained data from the California Cancer Registry linked to California Medicaid claims (CCR-M) for 1,406 stage I NSCLC cases who underwent definitive surgical management (as indicated in registry data)

and were diagnosed between 2007 and 2013. Using linked Medicaid claims we identified patients with COPD and CAD using relevant diagnostic codes (see above). We then generated survival curves, stratified by age group and comorbidity status, calculating 10-year overall survival with 95% confidence intervals. These were then compared to model output.

**Base case and sensitivity analyses.** For the base case analysis, we compared the effectiveness of lobectomy, limited resection, and SBRT. We used the LCPM-Treatment to estimate the life expectancy and quality adjusted life expectancy (QALE) of Veterans with stage I NSCLC for various permutations of treatment options, comorbid illness (categorizing COPD Gold Stage: 0–1,2, 3 and CAD as a binary status), age (<60, 60–69, 70–79, and ≥80), histology (adenocarcinoma vs. squamous cell carcinoma [SCC]), and tumor size (<1, 1–2, 2–3, and >3 cm). GOLD Stage 4 was not included in the simulation as it was considered unlikely that these patients would be considered surgical candidates. We selected optimal treatments for each permutation by identifying treatments that were associated with a ≥3 quality-adjusted month life gain, based on published criteria for clinical significance for oncologic interventions [31]. If treatments were not associated with a ≥3 gain in comparison to other choices, multiple optimal treatments were chosen.

We performed probabilistic sensitivity analysis to address the uncertainties in our model parameters for treatment efficacy. The upper and lower bounds associated with base case treatment efficacy estimates (Table 1) were based on the 95% confidence intervals associated with those treatment effects.

## Results

### Lung cancer parameter estimates

Using data from a large cohort of Veterans who underwent lung cancer surgery, we found that age >80 years, CAD and COPD were associated with an increased risk of 30-day mortality after lung cancer surgery (S1 and S2 Tables). History of CAD and COPD were also both associated with increased risks of major post-operative complications; CAD was associated with a nearly four-fold increased odds of post-operative MI (odds ratio [OR] 3.93; 95% confidence interval [CI]: 1.81–8.57). Diagnosis of COPD was associated with a number of major complications including atrial fibrillation, MI, pneumonia, reoperation, sepsis, bleeding, prolonged hospital stay, recurrent respiratory failure, reintubation and renal failure. In a model including information on severity of airway obstruction by GOLD status, increasing airway obstruction was associated with increased odds of prolonged hospital stay and reintubation. We also estimated the incidence of severe toxicity associated with SBRT; major toxicity was rare (S5 Table; 1.55%). The most common adverse events were esophagitis (0.77%), pneumonitis (0.52%) and hemoptysis (0.26%).

We fitted regression models of non-lung cancer mortality to estimate the effects of competing risks from age and comorbidities. Risk of non-lung cancer mortality was higher with increasing age (S3 Table; subhazard ratio [SHR]:1.70; 95%CI:1.10–2.61 for ages 60–69; HR: 3.07; 95%CI: 2.00–4.70 for ages 70–79 and HR: 5.00; 95%CI: 3.04–8.20 for ages ≥80). Conversely, CAD diagnosis was not significantly associated with an increased risk of non-lung cancer death (HR: 1.15; 95%CI: 0.71–1.85). For lung cancer-specific mortality, increasing age was associated with modest increases in lung cancer-related death (SHR: 1.39; 95% CI: 1.15–1.68 for ages 70–79, and SHR: 1.41; 95% CI: 1.04–1.91 for ages ≥80). Adenocarcinoma histology was associated with lower cancer-specific death (SHR 0.87; 95% CI: 0.76–0.99) and COPD was also associated with higher rates of lung cancer-specific death (SHR 1.20; 95% 1.05–1.38).

Using data from the VACS, we estimated the baseline utility decrement associated with major comorbidities. We found that COPD and CAD diagnosis were significantly associated with a disutility of -0.021 (S4 Table; p<0.001) and -0.018 (p = 0.005), respectively.

For external validation we compared 10-year survival estimates by age and comorbidity group to values from CCR-M cases. Our estimates fell within the 95% confidence intervals for 10-year survival for 76% of all simulations.

### Simulation models comparing early stage lung cancer treatments

We compared quality-adjusted life year (QALY) gains for major treatment modalities for stage I NSCLC for Veterans by estimating outcomes associated with each therapy in identical cohorts (Table 2). Although primary parameter analyses found that lobectomy had an increased risk of several major short-term surgical complications, it was still associated with the largest QALYs gained in the majority of scenarios.

**Veterans without CAD.** For all Veterans <80 years without CAD, lobectomy was the treatment associated with the greatest projected QALY gained; however, the magnitude of QALYs gained varied with tumor size (S6A–S6C Table). Veterans <80 years old (with or without COPD) with <1cm tumors had a 3–8% absolute QALY increase from lobectomy over SBRT or limited resection while the QALY increases of lobectomy for both histologic subtypes in the <80 age groups with larger tumors ranged from 8% to 16%. No single dominant treatments were identified for Veterans >80 years without CAD; for smaller tumors dominant options were less invasive treatments but lobectomy had similar projected QALYs gained to other modalities for tumors >1 cm.

**Veterans with CAD.** Optimal treatments for Veterans with COPD and CAD differed in several scenarios. For small tumors (<2 cm) of both histologic subtypes, lobectomy had the greatest projected QALYs gained compared to other treatments for Veterans with CAD for younger patients (<70 years of age). Absolute 10-year survival increases were greatest (up to 16%) for younger Veterans (age <60) with <2 cm tumors when comparing lobectomy to the projections for the treatment with the second highest QALYs gained. For Veterans with larger tumors (>2 cm) of both histologic subtypes, with comorbid CAD treatments with the highest projected QALYs gained varied and often several treatments were similar. In general less invasive treatments (limited resection or SBRT) tended to be associated with more QALYs gained in older patients, although in some situations the model found no clinically significant difference between the three treatment approaches.

The model inputs included the potential harms associated with each stage I NSCLC treatment modality which we stratified according to age and presence of CAD and/or COPD (Table 3). Pneumonia was the most prevalent complication of lung resection surgery, occurring in 12–17% (according to age group) of Veterans with either CAD or COPD who underwent lobectomy and 7–10% of those with comorbidities who underwent limited lung resection. Complications of SBRT were infrequent and did not vary according to age or comorbidity.

We evaluated the robustness of the model output by conducting sensitivity analyses, varying the estimates of treatment efficacy of the three modalities over their confidence ranges (S1 and S2 Figs). For scenarios where lobectomy was associated with the greatest projected QALE in the base-case it was typically the treatment with the highest QALEs gained in the majority of simulations across patient groups.

## Discussion

We extended an established microsimulation model to evaluate the impact of comorbidity on NSCLC treatment outcomes including life expectancy and QALE in Veterans. We found that

**Table 2. Non-small cell lung cancer stage I treatments that maximize quality-adjusted life year gains by comorbidity status\*.**

| Size / Histologic Subtype | Age Group (Years) | Optimal Treatment No Coronary Artery Disease | | | Optimal Treatment Coronary Artery Disease | | |
|---|---|---|---|---|---|---|---|
| | | GOLD 0–1 | GOLD 2 | GOLD 3 | GOLD 0–1 | GOLD 2 | GOLD 3 |
| <1 cm Adenocarcinoma | <60 | Lob | Lob | Lob | Lob | Lob | Lob |
| | 60–69 | Lob | Lob | Lob | Lob | Lob | Lob |
| | 70–79 | Lob | Lob | Lob | Lob | Lob | Lob/LR/SBRT |
| | ≥80 | LR/SBRT | LR/SBRT | LR/SBRT | LR/SBRT | LR/SBRT | LR/SBRT |
| 1–2 cm Adenocarcinoma | <60 | Lob | Lob | Lob | Lob | Lob | Lob |
| | 60–69 | Lob | Lob | Lob | Lob | Lob | Lob |
| | 70–79 | Lob | Lob | Lob | Lob | Lob | Lob |
| | ≥80 | LR/Lob/SBRT | LR/Lob/SBRT | LR/Lob/SBRT | LR/SBRT | LR/SBRT | LR/SBRT |
| 2–3 cm Adenocarcinoma | <60 | Lob | Lob | Lob | Lob | Lob | Lob |
| | 60–69 | Lob | Lob | Lob | Lob | Lob | Lob |
| | 70–79 | Lob | Lob | Lob | Lob | Lob | Lob |
| | ≥80 | LR/Lob/SBRT | LR/Lob/SBRT | LR/Lob/SBRT | LR/SBRT | LR/SBRT | LR/SBRT |
| >3cm Adenocarcinoma | <60 | Lob | Lob | Lob | Lob | Lob | Lob |
| | 60–69 | Lob | Lob | Lob | Lob | Lob | Lob |
| | 70–79 | Lob | Lob | Lob | Lob | Lob | Lob |
| | ≥80 | LR/Lob | LR/Lob | LR/Lob | LR | LR | LR |
| <1 cm Squamous Cell Carcinoma | <60 | Lob | Lob | Lob | Lob | Lob | Lob |
| | 60–69 | Lob | Lob | Lob | Lob | Lob | Lob |
| | 70–79 | Lob | Lob | Lob | Lob/LR/SBRT | Lob/LR/SBRT | Lob/LR/SBRT |
| | ≥80 | LR/SBRT | LR/SBRT | LR/SBRT | LR/SBRT | LR/SBRT | LR/SBRT |
| | | | | | 8.5 | | |
| 1–2 cm Squamous Cell Carcinoma | <60 | Lob | Lob | Lob | Lob | Lob | Lob |
| | 60–69 | Lob | Lob | Lob | Lob | Lob | Lob |
| | 70–79 | Lob | Lob | Lob | Lob | Lob | Lob |
| | ≥80 | LR/Lob/SBRT | LR/Lob/SBRT | LR/Lob/SBRT | LR/Lob/SBRT | LR/Lob/SBRT | LR/Lob/SBRT |
| | | | | | 8.5 | | |
| 2–3 cm Squamous Cell Carcinoma | <60 | Lob | Lob | Lob | Lob | Lob | Lob |
| | 60–69 | Lob | Lob | Lob | Lob | Lob | Lob |
| | 70–79 | Lob | Lob | Lob | Lob | Lob | Lob |
| | ≥80 | LR/Lob/SBRT | LR/Lob/SBRT | LR/Lob/SBRT | LR/Lob/SBRT | LR/Lob/SBRT | LR/Lob/SBRT |
| | | | | | 8.5 | 8. | |
| >3 cm Squamous Cell Carcinoma | <60 | Lob | Lob | Lob | Lob | Lob | Lob |
| | 60–69 | Lob | Lob | Lob | Lob | Lob | Lob |
| | 70–79 | Lob | Lob | Lob | Lob | Lob | Lob |
| | ≥80 | LR/Lob | LR/Lob | LR/Lob | LR/Lob | LR/Lob | LR/Lob |

LR: Limited resection; Lob: Lobectomy;

\*Treatment approach with >3 quality-adjusted month gain, for scenarios where <3 month gain, multiple approaches included.

projected outcomes for stage I tumor treatment strategies differed according to patient age, tumor characteristics, COPD severity and presence of CAD. For most Veterans lobectomy was associated with the largest projected QALYs gained; however, older patients with smaller tumors often had the highest projected QALYs gained from less extensive treatment approaches.

**Table 3. Probability of major treatment complications.**

| Treatment Complications | Age Group (Years) | Probability of Complication | | | | | | | | | | | |
|---|---|---|---|---|---|---|---|---|---|---|---|---|---|
| | | No CAD or COPD | | | CAD, no COPD | | | COPD, no CAD | | | CAD and COPD | | |
| | | L | LR | SBRT | L | LR | SBRT | L | LR | SBRT | L | LR | SBRT |
| | | | | | | | | | | | | | - |
| ARDS | <60 | 3.80% | 2.03% | - | 3.80% | 2.03% | - | 3.80% | 2.03% | - | 3.80% | 2.03% | - |
| | 60–69 | 3.80% | 2.03% | - | 3.80% | 2.03% | - | 3.80% | 2.03% | - | 3.80% | 2.03% | - |
| | 70–79 | 3.80% | 2.03% | - | 3.80% | 2.03% | - | 3.80% | 2.03% | - | 3.80% | 2.03% | - |
| | ≥80 | 3.80% | 2.03% | - | 3.80% | 2.03% | - | 3.80% | 2.03% | - | 3.80% | 2.03% | - |
| CVA | <60 | 0.38% | 0.20% | - | 0.38% | 0.20% | - | 0.75% | 0.39% | - | 0.75% | 0.39% | - |
| | 60–69 | 0.66% | 0.34% | - | 0.66% | 0.34% | - | 1.30% | 0.69% | - | 1.30% | 0.69% | - |
| | 70–79 | 1.15% | 0.61% | - | 1.15% | 0.61% | - | 2.27% | 1.21% | - | 2.27% | 1.21% | - |
| | ≥80 | 2.12% | 1.12% | - | 2.12% | 1.12% | - | 4.14% | 2.21% | - | 4.14% | 2.21% | - |
| Empyema | <60 | 0.84% | 0.44% | - | 0.84% | 0.44% | - | 1.36% | 0.71% | - | 1.36% | 0.71% | - |
| | 60–69 | 0.84% | 0.44% | - | 0.84% | 0.44% | - | 1.36% | 0.71% | - | 1.36% | 0.71% | - |
| | 70–79 | 0.84% | 0.44% | - | 0.84% | 0.44% | - | 1.36% | 0.71% | - | 1.36% | 0.71% | - |
| | ≥80 | 0.84% | 0.44% | - | 0.84% | 0.44% | - | 1.36% | 0.71% | - | 1.36% | 0.71% | - |
| MI | <60 | 0.90% | 0.37% | - | 3.44% | 1.44% | - | 1.74% | 0.72% | - | 6.53% | 2.78% | - |
| | 60–69 | 0.90% | 0.37% | - | 3.44% | 1.44% | - | 1.74% | 0.72% | - | 6.53% | 2.78% | - |
| | 70–79 | 0.90% | 0.37% | - | 3.44% | 1.44% | - | 1.74% | 0.72% | - | 6.53% | 2.78% | - |
| | ≥80 | 0.90% | 0.37% | - | 3.44% | 1.44% | - | 1.74% | 0.72% | - | 6.53% | 2.78% | - |
| Pneumonia | <60 | 6.57% | 3.76% | - | 6.57% | 3.76% | - | 12.23% | 7.18% | - | 12.23% | 7.18% | - |
| | 60–69 | 7.22% | 4.13% | - | 7.22% | 4.13% | - | 13.35% | 7.87% | - | 13.35% | 7.87% | - |
| | 70–79 | 7.91% | 4.55% | - | 7.91% | 4.55% | - | 14.55% | 8.63% | - | 14.55% | 8.63% | - |
| | ≥80 | 9.48% | 5.49% | - | 9.48% | 5.49% | - | 17.19% | 10.33% | - | 17.19% | 10.33% | - |
| Reoperation | <60 | 3.52% | 2.74% | - | 4.10% | 3.19% | - | 5.22% | 4.08% | - | 6.06% | 4.74% | - |
| | 60–69 | 3.52% | 2.74% | - | 4.10% | 3.19% | - | 5.22% | 4.08% | - | 6.06% | 4.74% | - |
| | 70–79 | 3.52% | 2.74% | - | 4.10% | 3.19% | - | 5.22% | 4.08% | - | 6.06% | 4.74% | - |
| | ≥80 | 3.52% | 2.74% | - | 4.10% | 3.19% | - | 5.22% | 4.08% | - | 6.06% | 4.74% | - |
| Respiratory Failure | <60 | 0.61% | 0.43% | - | 0.61% | 0.43% | - | 1.28% | 0.90% | - | 1.28% | 0.90% | - |
| | 60–69 | 0.75% | 0.53% | - | 0.75% | 0.53% | - | 1.56% | 1.10% | - | 1.56% | 1.10% | - |
| | 70–79 | 0.91% | 0.64% | - | 0.91% | 0.64% | - | 1.89% | 1.34% | - | 1.89% | 1.34% | - |
| | ≥80 | 1.05% | 0.74% | - | 1.05% | 0.74% | - | 2.17% | 1.54% | - | 2.17% | 1.54% | - |
| Sepsis | <60 | 3.02% | 1.74% | - | 3.02% | 1.74% | - | 5.86% | 3.42% | - | 5.86% | 3.42% | - |
| | 60–69 | 3.02% | 1.74% | - | 3.02% | 1.74% | - | 5.86% | 3.42% | - | 5.86% | 3.42% | - |
| | 70–79 | 3.02% | 1.74% | - | 3.02% | 1.74% | - | 5.86% | 3.42% | - | 5.86% | 3.42% | - |
| | ≥80 | 3.02% | 1.74% | - | 3.02% | 1.74% | - | 5.86% | 3.42% | - | 5.86% | 3.42% | - |
| Renal Failure | <60 | 1.32% | 0.70% | - | 1.32% | 0.70% | - | 4.32% | 2.31% | - | 4.32% | 2.31% | - |
| | 60–69 | 1.32% | 0.70% | - | 1.32% | 0.70% | - | 4.32% | 2.31% | - | 4.32% | 2.31% | - |
| | 70–79 | 1.32% | 0.70% | - | 1.32% | 0.70% | - | 4.32% | 2.31% | - | 4.32% | 2.31% | - |
| | ≥80 | 1.32% | 0.70% | - | 1.32% | 0.70% | - | 4.32% | 2.31% | - | 4.32% | 2.31% | - |
| Esophagitis | <60 | - | - | 0.77% | - | - | 0.77% | - | - | 0.77% | - | - | 0.77% |
| | 60–69 | - | - | 0.77% | - | - | 0.77% | - | - | 0.77% | - | - | 0.77% |
| | 70–79 | - | - | 0.77% | - | - | 0.77% | - | - | 0.77% | - | - | 0.77% |
| | ≥80 | - | - | 0.77% | - | - | 0.77% | - | - | 0.77% | - | - | 0.77% |
| Pneumonitis | <60 | - | - | 0.52% | - | - | 0.52% | - | - | 0.52% | - | - | 0.52% |
| | 60–69 | - | - | 0.52% | - | - | 0.52% | - | - | 0.52% | - | - | 0.52% |
| | 70–79 | - | - | 0.52% | - | - | 0.52% | - | - | 0.52% | - | - | 0.52% |
| | ≥80 | - | - | 0.52% | - | - | 0.52% | - | - | 0.52% | - | - | 0.52% |

(*Continued*)

**Table 3.** (Continued)

| Treatment Complications | Age Group (Years) | Probability of Complication | | | | | | | | | | | |
|---|---|---|---|---|---|---|---|---|---|---|---|---|---|
| | | No CAD or COPD | | | CAD, no COPD | | | COPD, no CAD | | | CAD and COPD | | |
| | | L | LR | SBRT | L | LR | SBRT | L | LR | SBRT | L | LR | SBRT |
| Hemoptysis | <60 | - | - | 0.26% | - | - | 0.26% | - | - | 0.26% | - | - | 0.26% |
| | 60–69 | - | - | 0.26% | - | - | 0.26% | - | - | 0.26% | - | - | 0.26% |
| | 70–79 | - | - | 0.26% | - | - | 0.26% | - | - | 0.26% | - | - | 0.26% |
| | ≥80 | - | - | 0.26% | - | - | 0.26% | - | - | 0.26% | - | - | 0.26% |

There is very limited guidance regarding optimal treatment pathways for stage I NSCLC patients that consider the impact of major comorbid illnesses on treatment complications, life expectancy and quality of life. Lacking randomized trial data in this patient population, the simulation approach employed in this study incorporated the best available evidence regarding treatment efficacy of the modalities (from VA research when possible) and used primary data from large Veteran cohorts to generate "in-silico" comparative trial data for these treatments. American College of Chest Physicians guidelines for the evaluation of lung cancer patients being considered for surgery advocate risk stratification with pulmonary and functional testing, but do not provide discrete recommendations for specific treatment approaches based on the results of these testing [32]. These guidelines also discuss cardiac risk stratification prior to the consideration of surgery, but do not provide lung cancer treatment recommendations based on the findings of that evaluation. Future analyses of the prospective Society of Thoracic Surgeons' General Thoracic Surgery Database are likely to provide additional prognostic information related to these risk factors [33]. Our simulation model provides quantitative estimates of treatment benefits by balancing the treatment effectiveness against the short-term morbidity and mortality associated with treatments and risk of non-lung cancer death. In most scenarios, the effectiveness of lobectomy outweighed the harms (that affected both life expectancy and quality of life within the simulations) but in select situations, less invasive treatments (such as sublobar resection) were associated with the greatest projected high-quality life expectancy gains for patients with small tumors, comorbidity, and/or advanced age.

The use of sublobar resection instead of lobectomy for NSCLC in patients with COPD and potential significant airway obstruction has been controversial. The only randomized trial comparing these approaches was not designed to evaluate this patient subgroup and did not have adequate power to compare outcomes for these patients [13]. A study incorporating national data from the National Surgical Quality Improvement Project included COPD as a major input in a risk score for identifying candidates for lobectomy versus a sublobar approach; however, this analysis only considered short-term surgical outcomes [34]. In contrast, several surgical series have shown that post-operative declines in lung function for patients with moderate to severe COPD undergoing lobectomy may not be large, providing some support for the use of more extensive resection in these patients [35]. Our simulation incorporated increased risks of major short-term complications associated the use of lobectomy versus limited resection and further considered the parameter of reduced lung function on non-lung cancer mortality. Despite this, our model output indicated that simulated patients younger than 80 years with moderate to severe COPD and no CAD still benefited most from lobectomy, although the projected QALY gained compared to limited lung resection was only modest. Thus, COPD patients that are risk averse to the potential complications of lobectomy may prefer limited resection.

Lung cancer patients with CAD, as ascertained by diagnostic codes, also differed in their optimal treatment. For small tumors (<1 cm) SBRT or limited lung resection was associated with the largest projected QALY gained due to the impact of greater surgical morbidity and resultant quality-of-life detriment from lobectomy as well as overall impact of CAD on life expectancy. Otherwise CAD had relatively limited impact on the optimal stage I NSCLC treatment regimens in our simulations. This is consistent with surgical series and cohort studies that have demonstrated greater perioperative morbidity and mortality for patients with CAD [36, 37]. As there are no explicit guidelines regarding treatment selection for stage I NSCLC for patients with CAD, these results consolidate relevant observational data to provide estimates of potential post-operative complications and QALE associated with different treatment strategies.

The major strengths of our study include using patient-level data from several large contemporary cohorts of Veterans and a well-established microsimulation model framework that has been previously used for numerous high impact lung cancer screening and treatment analyses. Our findings also have some limitations, however. First, our comorbidity information was informed by diagnostic codes from electronic medical records. We used spirometry data, however, to confirm COPD stage for most of our analyses, and diagnostic codes for CAD have been found to be highly specific for the presence of disease in validation studies [38]. Additionally, we incorporated estimates of early-stage lung cancer treatment efficacy using available evidence, but it is acknowledged that there are still knowledge gaps in the definitive comparison of these treatments. The superiority of lobectomy to limited lung resection was established in a single trial from the 1980-90s, prior to the widespread use of preoperative PET assessment (and lacking a requirement for full-body computed tomography staging) as well as video-assisted thorascopic techniques. Furthermore, we relied on observational data comparing SBRT to surgical techniques, as no high-quality randomized trial comparing the efficacy of SBRT to surgical modalities has been completed. To address these uncertainties we incorporated sensitivity analyses varying the effects of these treatments over their plausible ranges. Despite these limitations, this analysis still attempts to comprehensively consolidate all available data to create granular, discrete comparative effectiveness estimates for these treatments in a highly prevalent group of patients. Despite this, as these data are heavily reliant on observational data they should be primarily used to guide future research and not to guide clinical decisions. However, our simulation model can be rapidly updated once new randomized trial evidence regarding the treatment of early-stage NSCLC is published.

In summary, using a simulation model of treatment of early-stage lung cancer in Veterans with comorbid illnesses we found that lobectomy was associated with higher projected quality-adjusted life year gains in many patients; however, certain patient groups had very limited added benefit from more invasive approaches. These results can be used to inform future research in the treatment of lung cancer in Veterans with COPD and/or CAD, two common comorbidities in these patients.

## Supporting information

**S1 Table. Multivariable logistic regression models of 30-day complications of lung cancer surgery among veterans; Model 1 with COPD as a single covariate.**
(DOCX)

**S2 Table. Multivariable logistic regression models of 30-day complications of lung cancer surgery among veterans; Model 2 with GOLD stages of airway obstruction included.**
(DOCX)

**S3 Table. Multivariable regression of non-lung cancer death.**
(DOCX)

**S4 Table. Baseline quality of life (utility) values from the veterans aging cohort status according to comorbidity.**
(DOCX)

**S5 Table. Prevalence of major toxicity following SBRT treatment.**
(DOCX)

**S6 Table.** A. Estimates of quality-adjusted life year gains for different stage I NSCLC treatment options in veterans for patients with no COPD or GOLD stage 1. B. Estimates of quality-adjusted life year gains for different stage I NSCLC treatment options in veterans for patients with GOLD stage 2. C. Estimates of quality-adjusted life year gains for different stage I NSCLC treatment options in veterans for patients with GOLD stage 3.
(DOCX)

**S1 Fig. Probabilistic sensitivity analysis of optimal stage I NSCLC treatment strategies for adenocarcinoma histologic subtype.** Probabilities represent the proportion of simulations where a treatment strategy was the optimal modality for maximizing QALYs gained.
(TIF)

**S2 Fig. Probabilistic sensitivity analysis of optimal stage I NSCLC treatment strategies for squamous cell carcinoma histologic subtype.**
(TIF)

## Author Contributions

**Conceptualization:** Keith Sigel, Chung Yin Kong.

**Data curation:** Keith Sigel, Sadiq Rehmani, Yeun-Hee Park, Juan Wisnivesky.

**Formal analysis:** Keith Sigel, Chung Yin Kong, Sadiq Rehmani, Kimberly Stone, Minal Kale, Juan Wisnivesky.

**Funding acquisition:** Keith Sigel, Chung Yin Kong, Susan Bates, Juan Wisnivesky.

**Investigation:** Keith Sigel, Chung Yin Kong, Sadiq Rehmani, Susan Bates, Minal Kale, Kristina Crothers, Faiz Bhora, Juan Wisnivesky.

**Methodology:** Keith Sigel, Susan Bates, Michael Gould, Kimberly Stone, Minal Kale, Yeun-Hee Park, Kristina Crothers, Faiz Bhora, Juan Wisnivesky.

**Project administration:** Keith Sigel, Chung Yin Kong, Yeun-Hee Park, Juan Wisnivesky.

**Resources:** Keith Sigel.

**Software:** Chung Yin Kong.

**Supervision:** Keith Sigel.

**Validation:** Keith Sigel, Kimberly Stone.

**Writing – original draft:** Keith Sigel.

**Writing – review & editing:** Chung Yin Kong, Sadiq Rehmani, Susan Bates, Michael Gould, Kimberly Stone, Minal Kale, Yeun-Hee Park, Kristina Crothers, Faiz Bhora, Juan Wisnivesky.

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
