## [Decision Letter · Decision Letter 0]

18 Dec 2020

PONE-D-20-35238

Optimal Treatment Strategies for Stage I Non-small Cell Lung Cancer in Veterans with Pulmonary and Cardiac Comorbidities

PLOS ONE

Dear Dr. Sigel,

Thank you for submitting your manuscript to PLOS ONE. After careful consideration, we feel that it has merit but does not fully meet PLOS ONE’s publication criteria as it currently stands. Therefore, we invite you to submit a revised version of the manuscript that addresses the points raised during the review process.

The reviewers have made suggestions for you to consider and perhaps implement. Please consider these suggestions thoughtfully; they are meant to be constructive.

We look forward to receiving your revised manuscript.

Kind regards,

Hyun-Sung Lee, M.D., Ph.D.

Academic Editor

PLOS ONE

Journal Requirements:

"Dr. Wisnivesky has received consulting honorarium from Sanofi, Glaxosmithkline and Banook and research grants from Sanofi and Quorum."

Reviewers' comments:

Reviewer's Responses to Questions

**Comments to the Author**

1. Is the manuscript technically sound, and do the data support the conclusions?

Reviewer #1: Partly

Reviewer #2: Yes

Reviewer #3: Yes

2. Has the statistical analysis been performed appropriately and rigorously? 

Reviewer #1: Yes

Reviewer #2: Yes

Reviewer #3: Yes

3. Have the authors made all data underlying the findings in their manuscript fully available?

Reviewer #1: No

Reviewer #2: Yes

Reviewer #3: Yes

4. Is the manuscript presented in an intelligible fashion and written in standard English?

Reviewer #1: Yes

Reviewer #2: Yes

Reviewer #3: Yes

5. Review Comments to the Author

Reviewer #1: This observational study utilized national VA data to simulate differences in outcomes for Veterans treated with surgery or SBRT according to age and comorbidities. The manuscript over-interprets the findings to suggest that these data can be used to guide clinical management. Recommendations are made below to present the data in a more appropriate and scientific manner. A key data input was missing (Boyer et al JTO 2017)

Abstract background: it isn’t possible to identify the “optimal management” from an observational dataset. Please state a more appropriate objective for this study that considers the limitations and discloses that its aim is to find associations with outcomes. It’s important that this paper does not assert itself as a guide for clinical practice and to be clear that it is only hypothesis-generating.

Abstract results: please use a more scientific term instead of “the best”. Ensure it reflects the finding of an association without suggesting causality. Next, provide more specific about what outcome measure was superior.

Abstract conclusion: please change “varies” to “may vary”. Next, the last sentence is unacceptable as the findings of this simulation modeling can, at most, guide future research study design.

Line 83: to avoid suggestions of causality, please change “worse” to “lower”

Line 93: please balance this statement with at least one reference that states the opposite (e.g. Robinson et al JTO 2013) and also studies that show OS might not be different with SBRT (e.g. Shirvani et al JAMA 2014) or limited resection (numerous)

Line 96: Please clarify that clinical trials data may be relevant for some Veterans, but not for all. As written, it suggests all Veterans have serious comorbidities.

Line 104: it’s unclear what is meant by “evaluate different therapeutic options”. Is this to predict outcomes from a simulated model? Or, to use a complex model to summarize outcomes that occurred in the past? This clarification will be appreciated by readers.

Line 119: please clarify what national VA sources were identified. Be specific as there are many, each with their respective limitations that have been published on. Clarify if the model pulled data from publications (as suggested in Table 1). Please clarify what domains from the VA’s CDW were accessed. Please disclose the authorizations obtained to access these non-publicly available data.

Line 130: Boyer et al JTO 2017 is a critical input that is missing here. That study, through 2010, identified over 400 pts treated with SBRT. Please clarify the reason for this discrepancy as this report only identified 386 pts though 2015.

Line 213: the tempered and scientific summary of results in this paragraph are appropriate and can serve as a model to summarize other findings to avoid over-interpretation elsewhere in the results section and discussion

Line 258: the term “benefits” suggests causality. Please substitute.

Line 287: statements about optimal treatments cannot be made from these observational data. One can

only summarize what covariates are associated with an outcome

Line 289: the term “maximized” suggests causality. Please use terms that clarify the discovery of associations.

Line 290: please substitute a more appropriate term for “benefited”

Line 295: the term “best” isn’t qualified. Please change or remove

Line 300: please acknowledge the value of the STS prospective database

Line 306: “led to maximal” suggests causality. Please change.

Line 320: please find a more suitable term than “benefited” to avoid suggestion of causality

Line 321: please find a more suitable term than “gain”

Line 325: the term “led” should be changed

Line 327: one cannot assess “impact” from an observational study. Please use a more appropriate term

Line 331: please substitute the language that suggests this report can “provide a clinical framework for clinical decision-making”

Line 334: this opening statement isn’t qualified until the data from Boyer et al is considered, including its supplemental tables and figures

Line 341: please find a more scientific adjective than “best”

Line 343: please list the most important limitations of LCSG study which were the lack of PET and no requirements for whole-body CT staging

Line 350: as this is a limitations paragraph, a final disclaimer should be stated that these data are useful to inform future study design and should not be used to guide clinical decisions

Line 353: please use language that is more scientific than “the superior treatment”. Summarize what outcome measure demonstrated higher values.

Line 354: this final sentence needs to be modified to avoid making a clinical recommendation

Please review Welp et al in Sem Thor CV Surg 2020

Please review the limitations of the Bryant et al paper that was written about here: https://www.healio.com/news/hematology-oncology/20171228/surgery-extends-survival-in-earlystage-lung-cancer

Reviewer #2: This is a well-balanced and comprehensive retrospective analysis of lung cancer treatment and CAD in veterans. The concepts are highly impactful and timely. No major concerns are identified with the authors approach.

Reviewer #3: COMMENTS TO THE AUTHOR

This is an interesting and well-written study describing the application of simulation modeling to evaluate optimal treatment of Veteran patients with stage I NSCLC with COPD and/or CAD. Study findings indicated that lobectomy was optimal for patient under 70 regardless of comorbidity status. While in general there was no optimal treatment for those over 80, treatment varied according to CAD status. This is a very timely and unique approach to investigating ideal treatment for stage I NSCLC for which SBRT is evolving as a recommended treatment option in certain situation. These findings will be helpful in clinical decision-making for Veterans, which is a population that’s older and has more comorbidities than the general population. Comments for consideration are provided below:

1. The authors used evidence from several Veteran-specific and other data sources incorporate estimates into the model. More detail is needed to help the reader understand the study populations the estimates were derived from in order to put into context the similarities/differences of the various cohorts. For example, the LC and non-LC mortality was based on CDW cancer registry data of 14, 029 patients diagnosed 2000-2015 with stage I-IIIA NSCLC retaining those who had lobectomy. So the lobectomy cohort was among those with stage I-IIIA and not just stage I? For the VASQIP--VA cancer registry linkage, were the 6,022 patients from the same diagnosis period and stage? Similarly, for the SBRT cohort, was this from the same pts diagnosed 2000-2015? For the VACS, what was the time period of data collection, especially since this was a non-cancer cohort. Additional clarification would be helpful to better understand the source of model inputs and potential impact on results.

2. Was there any consideration or ability to incorporate CAD severity (eg number of obstructive coronary arteries or other scoring system), as done for COPD, rather than just a binary variable?

3. It’s not clear if/how race/ethnicity was factored in the analyses? It would be important for the authors to discuss this as curative therapy has been shown to vary by race/ethnicity as well as significant race/ethnic differences in the major comorbidities (CAD, COPD) of interest in this study.

4. Minor comment: on Table 3, I think the first column is incorrectly labeled as ‘size/histologic subtype’ since it includes the treatment complications.

6. PLOS authors have the option to publish the peer review history of their article (what does this mean?). If published, this will include your full peer review and any attached files.

Reviewer #1: No

Reviewer #2: **Yes: **Farrah Kheradmand MD

Reviewer #3: No

---

## [Author Response · Author response to Decision Letter 0]

17 Feb 2021

Editor and Reviewer Response:

Response: We believe that we have met the style requirements. 

"Dr. Wisnivesky has received consulting honorarium from Sanofi, Glaxosmithkline and Banook and research grants from Sanofi and Quorum."

Please confirm that this does not alter your adherence to all PLOS ONE policies on sharing data and materials, by including the following statement: "This does not alter our adherence to PLOS ONE policies on sharing data and materials.” (as detailed online in our guide for authors [http://journals.plos.org/plosone/s/competing-interests<about:blank>](http://journals.plos.org/plosone/s/competing-interests%3Cabout:blank%3E) ). If there are restrictions on sharing of data and/or materials, please state these. Please note that we cannot proceed with consideration of your article until this information has been declared.

Response: We have updated the competing interests statement as suggested.

3. We note that you have indicated that data from this study are available upon request. PLOS only allows data to be available upon request if there are legal or ethical restrictions on sharing data publicly. For information on unacceptable data access restrictions, please see [http://journals.plos.org/plosone/s/data-availability#loc-unacceptable-data-access-restrictions](http://journals.plos.org/plosone/s/data-availability#loc-unacceptable-data-access-restrictions) < [https://urldefense.proofpoint.com/v2/url?u=http-3A__journals.plos.org_plosone_s_data-2Davailability-23loc-2Dunacceptable-2Ddata-2Daccess-2Drestrictions&d=DwQGaQ&c=shNJtf5dKgNcPZ6Yh64b-A&r=fzH-ffBjgLZBktYeWmbGZZLpAt-Zlr-osXvb9tKELHY&m=GqPpB2hazjn-kkNsmcF0YBaBIhXCjOEy9VkC1mU2pF4&s=_Zy8e2AwFVe66fDzr4l5U2gARbpnlPW7coZ-p1TAFRs&e=](https://urldefense.proofpoint.com/v2/url?u=http-3A__journals.plos.org_plosone_s_data-2Davailability-23loc-2Dunacceptable-2Ddata-2Daccess-2Drestrictions&d=DwQGaQ&c=shNJtf5dKgNcPZ6Yh64b-A&r=fzH-ffBjgLZBktYeWmbGZZLpAt-Zlr-osXvb9tKELHY&m=GqPpB2hazjn-kkNsmcF0YBaBIhXCjOEy9VkC1mU2pF4&s=_Zy8e2AwFVe66fDzr4l5U2gARbpnlPW7coZ-p1TAFRs&e=) >.

Response: We have revised the cover letter with the following:

The data used to create the simulation model in this study is legally protected Veterans Health Administration data that cannot be shared under any circumstances. Only aggregate data, as reported in this paper can be shared due to legal protections on Veterans health data. The data used in this project was under a waiver of informed consent granted by the Bronx VA Institutional Review Board and therefore even deidentified individual level data cannot be shared. https://www.va.gov/ORO/Docs/Guidance/VA_RSCH_DATA_ACCESS_PLAN_07_23_2015.pdf

The source data can be accessed in collaboration with VA researchers, free of charge with appropriate VA IRB approvals. Those wishing to access the raw data that were used for this analysis may contact Keith Sigel, MD PhD (keith.sigel@mssm.edu) to discuss the details of the VA data access approval process. We have confirmed that an interested researcher would be able to obtain a de-identified dataset upon request pending ethical approval. 

b) If there are no restrictions, please upload the minimal anonymized data set necessary to replicate your study findings as either Supporting Information files or to a stable, public repository and provide us with the relevant URLs, DOIs, or accession numbers. 

Response: See above. Individual level data cannot be removed or shared from our VA data source. 

4. Please include captions for your Supporting Information files at the end of your manuscript, and update any in-text citations to match accordingly. Please see our Supporting Information guidelines for more information: [http://journals.plos.org/plosone/s/supporting-information<about:blank>](http://journals.plos.org/plosone/s/supporting-information%3Cabout:blank%3E) .

Reviewers' comments:

5. Review Comments to the Author

Reviewer #1: This observational study utilized national VA data to simulate differences in outcomes for Veterans treated with surgery or SBRT according to age and comorbidities. The manuscript over-interprets the findings to suggest that these data can be used to guide clinical management. Recommendations are made below to present the data in a more appropriate and scientific manner. A key data input was missing (Boyer et al JTO 2017)

Abstract background: it isn’t possible to identify the “optimal management” from an observational dataset. Please state a more appropriate objective for this study that considers the limitations and discloses that its aim is to find associations with outcomes. It’s important that this paper does not assert itself as a guide for clinical practice and to be clear that it is only hypothesis-generating.

Response: This comment, and the overall emphasis on limiting the clinical applicability of our findings is appreciated. This statement has been revised with less strong language. 

Abstract results: please use a more scientific term instead of “the best”. Ensure it reflects the finding of an association without suggesting causality. Next, provide more specific about what outcome measure was superior.

Response: This has been revised.

Abstract conclusion: please change “varies” to “may vary”. Next, the last sentence is unacceptable as the findings of this simulation modeling can, at most, guide future research study design.

Response: This has been revised per reviewer suggestion. 

Line 83: to avoid suggestions of causality, please change “worse” to “lower”

Response: This has been changed per reviewer suggestion. 

Line 93: please balance this statement with at least one reference that states the opposite (e.g. Robinson et al JTO 2013) and also studies that show OS might not be different with SBRT (e.g. Shirvani et al JAMA 2014) or limited resection (numerous)

Response: These references and a modification of language have been added. 

Line 96: Please clarify that clinical trials data may be relevant for some Veterans, but not for all. As written, it suggests all Veterans have serious comorbidities.

Response: We have modified this language to address this comment. 

Line 104: it’s unclear what is meant by “evaluate different therapeutic options”. Is this to predict outcomes from a simulated model? Or, to use a complex model to summarize outcomes that occurred in the past? This clarification will be appreciated by readers.

Response: Thank you for this comment. We have revised this sentence. 

Line 119: please clarify what national VA sources were identified. Be specific as there are many, each with their respective limitations that have been published on. Clarify if the model pulled data from publications (as suggested in Table 1). Please clarify what domains from the VA’s CDW were accessed. Please disclose the authorizations obtained to access these non-publicly available data.

Response: These clarifications have been provided. 

Line 130: Boyer et al JTO 2017 is a critical input that is missing here. That study, through 2010, identified over 400 pts treated with SBRT. Please clarify the reason for this discrepancy as this report only identified 386 pts though 2015.

Response: We only included Veterans with lung cancer who had SBRT as indicated by Oncology file data; Boyer et al also used CPT codes – our opinion was that oncology file designation was the most accurate source (as it is collected by trained registrars). Please note that our SBRT cohort was only used to estimate complications of SBRT which were similar to other published estimates. Also, the treatment outcome differences in propensity score-matched analyses for SBRT versus lobectomy in Boyer et al were similar to the parameters that we used in our simulation (and our parameters were within their reported confidence intervals). 

Line 213: the tempered and scientific summary of results in this paragraph are appropriate and can serve as a model to summarize other findings to avoid over-interpretation elsewhere in the results section and discussion

Response: We have revised and tempered our language throughout this section. This reflects the next range of comments – from lines 258 - 327

Line 258: the term “benefits” suggests causality. Please substitute.

Response: This has been changed.

Line 287: statements about optimal treatments cannot be made from these observational data. One can only summarize what covariates are associated with an outcome

Response: We appreciate this comment and have removed language that might suggest cause. 

Line 289: the term “maximized” suggests causality. Please use terms that clarify the discovery of associations.

Response: As above, we have tempered this language. 

Line 290: please substitute a more appropriate term for “benefited”

Response: We have modified this to “associated with greatest projected QALE”

Line 295: the term “best” isn’t qualified. Please change or remove

Response: This is removed. 

Line 300: please acknowledge the value of the STS prospective database

Response: We have added an acknowledgement of the STS database. 

Line 306: “led to maximal” suggests causality. Please change.

Response: This has been changed. 

Line 320: please find a more suitable term than “benefited” to avoid suggestion of causality

Response: This has been changed.

Line 321: please find a more suitable term than “gain”

Response: This, as in many other locations, has been qualified as a projection – to clarify that this is not a causal conclusion. 

Line 325: the term “led” should be changed

Response: Have changed this to associational language. 

Line 327: one cannot assess “impact” from an observational study. Please use a more appropriate term

Response: This language has been changed. 

Line 331: please substitute the language that suggests this report can “provide a clinical framework for clinical decision-making”

Response: This has been removed. 

Line 334: this opening statement isn’t qualified until the data from Boyer et al is considered, including its supplemental tables and figures

Response: Please see above; the data from Boyer et al is consistent with our simulation parameters. 

Line 341: please find a more scientific adjective than “best”

Response: This has been removed. 

Line 343: please list the most important limitations of LCSG study which were the lack of PET and no requirements for whole-body CT staging

Response: This has been added.

Line 350: as this is a limitations paragraph, a final disclaimer should be stated that these data are useful to inform future study design and should not be used to guide clinical decisions

Response: This has been added, nearly verbatim.

Line 353: please use language that is more scientific than “the superior treatment”. Summarize what outcome measure demonstrated higher values.

Response: This has been edited.

Line 354: this final sentence needs to be modified to avoid making a clinical recommendation

Response: This has been edited. 

Reviewer #2: This is a well-balanced and comprehensive retrospective analysis of lung cancer treatment and CAD in veterans. The concepts are highly impactful and timely. No major concerns are identified with the authors approach.

Reponse: We thank the reviewer for these comments. 

Reviewer #3: COMMENTS TO THE AUTHOR

This is an interesting and well-written study describing the application of simulation modeling to evaluate optimal treatment of Veteran patients with stage I NSCLC with COPD and/or CAD. Study findings indicated that lobectomy was optimal for patient under 70 regardless of comorbidity status. While in general there was no optimal treatment for those over 80, treatment varied according to CAD status. This is a very timely and unique approach to investigating ideal treatment for stage I NSCLC for which SBRT is evolving as a recommended treatment option in certain situation. These findings will be helpful in clinical decision-making for Veterans, which is a population that’s older and has more comorbidities than the general population. Comments for consideration are provided below:

1. The authors used evidence from several Veteran-specific and other data sources incorporate estimates into the model. More detail is needed to help the reader understand the study populations the estimates were derived from in order to put into context the similarities/differences of the various cohorts. For example, the LC and non-LC mortality was based on CDW cancer registry data of 14, 029 patients diagnosed 2000-2015 with stage I-IIIA NSCLC retaining those who had lobectomy. So the lobectomy cohort was among those with stage I-IIIA and not just stage I? For the VASQIP--VA cancer registry linkage, were the 6,022 patients from the same diagnosis period and stage? Similarly, for the SBRT cohort, was this from the same pts diagnosed 2000-2015? For the VACS, what was the time period of data collection, especially since this was a non-cancer cohort. Additional clarification would be helpful to better understand the source of model inputs and potential impact on results.

Response: We have added these details to the methods; the years for all samples. All lung cancer cases were derived from the same source cohort in the corporate datawarehouse, identified in the VA oncology files 2000-2015. The participant data from VACS was from 2001-2006. 

2. Was there any consideration or ability to incorporate CAD severity (eg number of obstructive coronary arteries or other scoring system), as done for COPD, rather than just a binary variable?

Response: We did not have comprehensive information to estimate this risk and it was not included or considered for this analysis. 

3. It’s not clear if/how race/ethnicity was factored in the analyses? It would be important for the authors to discuss this as curative therapy has been shown to vary by race/ethnicity as well as significant race/ethnic differences in the major comorbidities (CAD, COPD) of interest in this study.

Response: We have intentionally omitted race/ethnicity as a factor in this model due to a concern that treatment disparities that are linked to race, ethnicity and sociodemographics may impact parameterization and subsequent model output might provide data that inadvertently amplifies disparities. 

4. Minor comment: on Table 3, I think the first column is incorrectly labeled as ‘size/histologic subtype’ since it includes the treatment complications.

Response: Thank you for this comment, we have revised this column heading.

---

## [Editor Report · Decision Letter 1]

19 Feb 2021

Optimal Treatment Strategies for Stage I Non-small Cell Lung Cancer in Veterans with Pulmonary and Cardiac Comorbidities

PONE-D-20-35238R1

Dear Dr. Sigel,

We’re pleased to inform you that your manuscript has been judged scientifically suitable for publication and will be formally accepted for publication once it meets all outstanding technical requirements.

Kind regards,

Hyun-Sung Lee, M.D., Ph.D.

Academic Editor

PLOS ONE
---

## [Editor Report · Acceptance letter]

8 Mar 2021

PONE-D-20-35238R1 

Optimal Treatment Strategies for Stage I Non-small Cell Lung Cancer in Veterans with Pulmonary and Cardiac Comorbidities 

Dear Dr. Sigel:

I'm pleased to inform you that your manuscript has been deemed suitable for publication in PLOS ONE. Congratulations! Your manuscript is now with our production department. 

Kind regards, 

on behalf of

Dr. Hyun-Sung Lee 

Academic Editor

PLOS ONE